# Investigating unmet need for healthcare using the European Health Interview Survey: a cross-sectional survey study of Luxembourg

Valerie Moran 🄳 ,[1,2] Marc Suhrcke,[3,4] Maria Ruiz-Castell,[1] Jessica Barré,[5] Laetitia Huiart[6]

► Prepublication history and additional online supplemental material for this paper are available online. To view these files, please visit the journal online (http://dx.doi.org/10.1136/bmjopen-2021-048860).

For numbered affiliations see end of article.

**Correspondence to**
Dr Valerie Moran;
valerie.moran@lih.lu

## ABSTRACT

**Objectives** We investigate the prevalence of unmet need arising from wait times, distance/transportation and financial affordability using the European Health Interview Survey. We explore associations between individual characteristics and the probability of reporting unmet need.

**Design** Cross-sectional survey conducted between February and December 2014.

**Setting and participants** 4004 members of the resident population in private households registered with the health insurance fund in Luxembourg aged 15 years and over.

**Outcome measures** Six binary variables that measured unmet need arising from wait time, distance/transportation and affordability of medical, dental and mental healthcare and prescribed medicines among those who reported a need for care.

**Results** The most common barrier to access arose from wait times (32%) and the least common from distance/transportation (4%). Dental care (12%) was most often reported as unaffordable, followed by prescribed medicines (6%), medical (5%) and mental health (5%) care. Respondents who reported bad/very bad health were associated with a higher risk of unmet need compared with those with good/very good health (wait: OR 2.41, 95% CI 1.53 to 3.80, distance/transportation: OR 7.12, 95% CI 2.91 to 17.44, afford medical care: OR 5.35, 95% CI 2.39 to 11.95, afford dental care: OR 3.26, 95% CI 1.86 to 5.71, afford prescribed medicines: OR 2.22, 95% CI 1.04 to 4.71, afford mental healthcare: OR 3.58, 95% CI 1.25 to 10.30). Income between the fourth and fifth quintiles was associated with a lower risk of unmet need for dental care (OR 0.29, 95% CI 0.16 to 0.53), prescribed medicines (OR 0.38, 95% CI 0.17 to 0.82) and mental healthcare (OR 0.17, 95% CI 0.05 to 0.61) compared with income between the first and second quintiles.

**Conclusions** Recent and planned reforms to address waiting times and financial barriers to accessing healthcare may help to address unmet need. In addition, policy-makers should consider additional policies targeted at high-risk groups with poor health and low incomes.

## INTRODUCTION

In recent years, the concept of universal health coverage (UHC) has emerged as

### Strengths and limitations of this study

► We investigated unmet need using a representative cross-sectional population based survey—the European Health Interview Survey (EHIS).
► The EHIS allowed us to investigate unmet need due to the affordability of prescribed medications and mental healthcare, in addition to medical and dental care.
► As unmet need was self-reported, it may be subject to recall bias or response bias.
► The EHIS does not collect data on unmet need due to wait for different types of services.
► This study explored associations between unmet need and sociodemographic, health status and risk factor variables but did not establish causality.

a key objective of international organisations including the WHO, Organisation for Economic Co-Operation and Development (OECD) and World Bank, who support national governments to achieve this goal. UHC is defined as 'ensuring that all people have access to needed health services…of sufficient quality to be effective while also ensuring that the use of these services does not expose the user to financial hardship'.[1] Therefore, access to care is a core component of UHC. At the European Union (EU) level, the European Parliament, Council and Commission announced the European Pillar of Social Rights in 2017.[2] The Pillar comprises 20 guiding principles that underpin a fair and inclusive society. Chapter 3 of the Pillar relates to social protection and inclusion and contains the principle that everyone is entitled to timely access to good quality affordable healthcare.[3] Barriers to accessing healthcare arising from the cost, physical accessibility and quality of services can lead to unmet need at an individual level due to affordability, distance and waiting times.[4 5]

Unmet need may also arise from other factors related to personal choices and circumstances including a lack of time to seek care due to family or work responsibilities, fear or dislike of medical personnel and treatment, a preference to wait and see if the symptoms resolve by themselves without seeking care and issues related to health literacy including language problems and a lack of knowledge of appropriate medical care.[6 7]

Countries have introduced various policies to address different aspects of unmet need. Many countries have implemented maximum waiting times, particularly for specialist consultations and elective treatments.[8] Countries have also introduced policies to reduce financial barriers to accessing healthcare. Belgium has put in place a range of financial protection measures including lower copayments for vulnerable groups and ceilings on the total amount of copayments paid by a household based on household income, regulation of supplementary payments and the third-party payment system (outlined in more detail below).[9] In France, individuals with chronic illnesses, pregnant women, low-income groups and individuals who suffered a work accident are exempt from most or all copayments.[10 11]

### Previous literature on unmet need for healthcare

Survey data are commonly used to ascertain individuals' perceptions of unmet need arising from various barriers to accessing care.[12 13] To date, studies that investigated unmet need within and across European countries have used data from the European Union Statistics on Income and Living Conditions (EU-SILC),[14–25] the European Social Survey (ESS)[26 27] and the European Health Interview Survey (EHIS).[28–30] Several studies that investigated unmet need for medical care[14 16 18 19 21 27] found that females were associated with higher unmet need compared with males. While there were conflicting results across studies for variables measuring age, education, employment, immigrant status and urban vs rural area, there was consensus for other covariates. Respondents in poorer health[14 16 23 25–27] and those with a chronic condition or illness[14 16 25 27] were more likely to report unmet need. Higher income groups were less likely to report unmet need[14 16 23 25 27] as were respondents with greater social capital and social support.[19 27] A small number of studies examined the determinants of unmet need for different services. Chaupain-Guillot and Guillot[15] found that older age was associated with a lower probability of reporting unmet need for medical and dental care while poorer health status and lower income was associated with a higher probability. However, while a higher level of education was associated with an increased probability of reporting unmet need for medical care, it was associated with a reduced probability of reporting unmet need for dental care. Hoebel et al[28] investigated unmet need among older people with low socioeconomic status in Germany. Among those aged 50–64, low socioeconomic status was associated with higher unmet need for medical, dental and mental healthcare for both men and women.

However, among the 65–85 age group low socioeconomic status was associated with higher unmet need only among men for medical and dental care. Rotarou and Sakellariou and Sakellariou and Rotarou[29 30] found that people with a disability were more likely to report higher unmet need for medical, dental and mental healthcare in Greece and the UK, respectively.

### The Luxembourgish health system and access to healthcare

Luxembourg provides universal coverage of healthcare through a mandatory social health insurance system, the Caisse Nationale de Santé (CNS). In 2018, the CNS covered 93% of the resident population.[31 32] The proportion not covered included EU officials based in Luxembourg, who were insured under a separate scheme provided by their employer[33] and vulnerable populations including the homeless and irregular immigrants and their families.[34]

In Luxembourg, when patients access healthcare including doctor consultations in outpatient or inpatient settings, dental and paramedical services, they must pay providers the full cost on receipt of care and then apply for a refund from the CNS for the covered share of the payment (excluding copayments). The CNS directly reimburses providers for hospital services (excluding the doctors' fees), laboratory tests and pharmaceuticals, leaving the patient to pay only the copayment at the point of use.[35]

Luxembourg is among a minority of countries (including Belgium and France) in the EU where patients pay ambulatory care providers directly and then claim reimbursement from the social health insurance fund.[36] The payment of the full cost of medical, dental and mental healthcare by the patient on receipt of care may create a financial barrier to accessing care for certain groups, for example, those on lower incomes.[34] In Belgium and France, specific population groups including those on low incomes or with a chronic illness have their costs directly covered by the social insurance fund, a scheme known as the 'tiers payant' or third-party payment system.[10 37] In Luxembourg, the government introduced the system of 'third-party social payment' ('tiers payant social') in 2013, which entitles people in economic hardship to request assistance with the payment of healthcare expenses.[38] Eligible patients are exempted from the payment of costs and the CNS reimburses the provider. The local social welfare office covers any copayments the patient cannot afford. The purpose of this policy is to enable people in economic hardship to access healthcare.[38] Therefore, the third-party payment system addresses unmet need arising from financial barriers to care in order to reduce financial hardship arising from healthcare utilisation.[9] However, it is unclear what the impact of this policy has been on addressing unmet need due to affordability of care in Luxembourg, as the policy has not been evaluated to date. Nevertheless, following a national debate over recent years, the government announced in November 2019 that a universal system of third-party payment would

be introduced,[39] which would cover the entire enrolled population of the CNS and replace the existing third-party social payment.

In addition to financial access to care, waiting times for health services have been identified as an important policy issue in Luxembourg. Waiting times are commonly a pertinent issue in countries with a national health system funded by general taxation. However, a recent report highlighted that waiting times were a medium to high priority in Luxembourg, in contrast to its neighbouring countries of Belgium and Germany, where waiting times were a low to medium or low priority.[8] This report also revealed that waiting times were an issue across different types of services including specialist care, diagnostic tests, hospital emergency departments, primary care and cancer care. Data on waiting times for healthcare in Luxembourg are not routinely published. However, a 2016 study revealed an average wait of almost 4 hours between admission to and discharge from emergency services, which are provided by each of the four general hospitals. The majority (75%) of attendees experienced a wait of 3 hours or less, just below the government target of 85%.[40] These findings prompted the government to set a maximum waiting time target of 2.5 hours for emergency services.[41] Efforts have also been undertaken to improve the organisation of cancer care in order to reduce waiting times. In 2016, the government embarked on a reform of the National Health Laboratory's diagnostic services by reducing outsourcing to other countries and concentrating the delivery of these services in Luxembourgish hospitals instead. The government also introduced maximum waiting time targets for cancer care. At least 95% of patients should receive a diagnosis within five working days while specific targets are in place for different types of cancer (eg, a maximum of 4 weeks between chemotherapy and radiotherapy or 2 weeks following receipt of the analytical report).[8] If residents of Luxembourg enrolled in the CNS perceive wait times as too long, they may seek healthcare in another EU state. Prior authorisation from the CNS is not required for a doctor consultation (in a health centre, clinic or hospital) but is required if the consultation uses specialised medical equipment or hospital infrastructure or for inpatient treatments with at least one overnight stay. The CNS may withhold authorisation if the necessary treatment can be provided in Luxembourg within a medically justifiable time frame.[42]

The issue of unmet need for healthcare in Luxembourg has not been studied to date despite its policy relevance and the availability of relevant survey data. The objective of this paper is to investigate the prevalence and determinants of unmet need in Luxembourg. We used EHIS data as it allowed us to explore the prevalence and determinants of unmet need separately for waiting time, distance or transportation and the affordability of medical, dental and mental healthcare and prescribed medicines. Therefore, we also contribute to the limited number of studies that investigated unmet need not only for medical care but also for dental and mental healthcare and prescribed medications.

## DATA AND METHODS
### Study population and design

The EHIS is a cross-sectional observational survey undertaken in all EU countries. A first wave of data was collected in 17 EU member states (excluding Luxembourg) between 2006 and 2009.[43] A second wave of the survey (EHIS 2) was collected in all 28 EU countries together with Iceland, Norway and Turkey between 2013 and 2015. Detailed information on the EHIS 2 methodology is available in a manual published by Eurostat[43] while a paper published by the Robert Koch Institute in Germany[44] provides a concise overview of the background and study methodology of the EHIS 2. Information on the EHIS two for Luxembourg, including the questionnaire and data access procedure is available on the Ministry of Health Directorate of Health website.[45] The survey collected information on health status, health determinants, utilisation of and barriers to access to healthcare and sociodemographic characteristics.[46] The coverage of the survey included the resident population in private households aged 15 years and over. A one-stage random sample stratified by age, sex and district of residence (Luxembourg, Diekirch and Grevenmacher) was drawn from the registry of CNS insurees.[45 47] Among the 16 000 individuals invited to participate, 4823 responded (response rate of 30.1%) by submitting an electronic (70%) or paper (30%) questionnaire.[46] Of these respondents, 4118 participants met the inclusion criteria, provided informed consent and completed the questionnaire (participation rate of 24.7%). Data were collected between February and December 2014.[47] The EHIS 2 Luxembourg database comprised 4004 individuals who completed more than 50% of the questionnaire and had no missing data for age, sex or district, (final participation rate of 25%). This database was prepared according to a European protocol[48] and was validated by Eurostat.[45]

The EHIS differs from the EU-SILC and ESS as it does not ask respondents a binary (yes/no) question on whether they have unmet need. Rather, the EHIS asks respondents to consider unmet need arising from specific barriers to accessing healthcare, including long waits, distance or transportation problems and the affordability of services. The EHIS data allows the investigation of each component of unmet need separately and the consideration of financial barriers for medical, dental and mental healthcare and prescribed medicines. The survey questions are available in online supplemental appendix 1. Table 1 shows the number and percentage of respondents who reported no unmet need, an unmet need or no need for healthcare for each component of unmet need. The percentage of respondents who reported no need for healthcare ranged from 15% for affordability of dental care and prescribed medicines to 38% for mental healthcare.

**Table 1** Responses for the six components of unmet need

| | Wait | | Distance or transportation problems | | Could not afford medical care | | Could not afford dental care | | Could not afford prescribed medicines | | Could not afford mental healthcare | |
|---|---|---|---|---|---|---|---|---|---|---|---|---|
| | N | % | N | % | N | % | N | % | N | % | N | % |
| No | 1906 | 48 | 2482 | 62 | 2866 | 72 | 2809 | 70 | 2977 | 74 | 2097 | 52 |
| Yes | 878 | 22 | 92 | 2 | 182 | 5 | 399 | 10 | 216 | 5 | 106 | 3 |
| No need for healthcare | 1043 | 26 | 983 | 25 | 782 | 20 | 599 | 15 | 609 | 15 | 1533 | 38 |
| Missing | 177 | 4 | 447 | 11 | 174 | 4 | 197 | 5 | 202 | 5 | 268 | 7 |
| Total | 4004 | 100 | 4004 | 100 | 4004 | 100 | 4004 | 100 | 4004 | 100 | 4004 | 100 |

## Outcome variables

For our outcome variables, we created six binary variables to measure barriers to access due to wait time, distance and affordability of medical care, dental care, prescribed medicines and mental healthcare among those who reported a need for care. We coded these variables as one, if respondents replied 'yes', as zero if respondents replied 'no' and as missing if respondents reported 'no need for healthcare'.

## Explanatory variables

Explanatory variables covered sociodemographics, health behaviours and health status. Sociodemographic variables included sex, age, marital status, immigrant, education, employment status, income, social support and being an informal carer. Sex, marital status, immigrant and informal carer were constructed as binary variables while age, education and employment status were constructed as categorical variables. Household income was measured in quintiles and included as a categorical variable. Following Ruiz-Castell *et al*,[49] we created a categorical variable on low, moderate and high social support using the questions: 'how much concern do people show in what you are doing?' and 'how easy is it to get practical help from neighbours if you should need it?'. Health risk factors included body mass index (BMI), smoking and alcohol consumption. We considered alcohol consumption as 'irregular', if it occurred at most 2–3 days per month, and 'regular', if it occurred at least once per week. We used three measures of health status: self-assessed health, presence of a chronic disease and limitations in activities due to health problems. We also included binary variables (fixed effects) for the twelve cantons (see figure 1) to capture geographical differences in unmet need.

We excluded observations with missing data for the explanatory variables. Online supplemental appendix 1, table A1 shows the percentage of missing data for each explanatory variable. Missing data for the explanatory variables did not exceed 7%, except for income, where 29% of observations had missing data. Online supplemental appendix 1, table A2 shows the characteristics of respondents with missing income data. These were more likely female, aged 15–24, unmarried, of Luxembourgish nationality, had only primary and pre-primary education, were students, fulfilling domestic tasks or in compulsory

service, had no limitations in activities due to a health problem and were less likely to be overweight or smoke daily.

## Statistical data analysis

We conducted separate analyses for each component of unmet need and investigated associations between the outcome and explanatory variables using multivariate logistic regression models of the following form:

$$P(y = 1 \mid x) = \frac{e^{\alpha + \beta x}}{1 + e^{(\alpha + \beta x)}}$$

where y is unmet need due to wait, distance or affordability and X is a vector of explanatory variables.

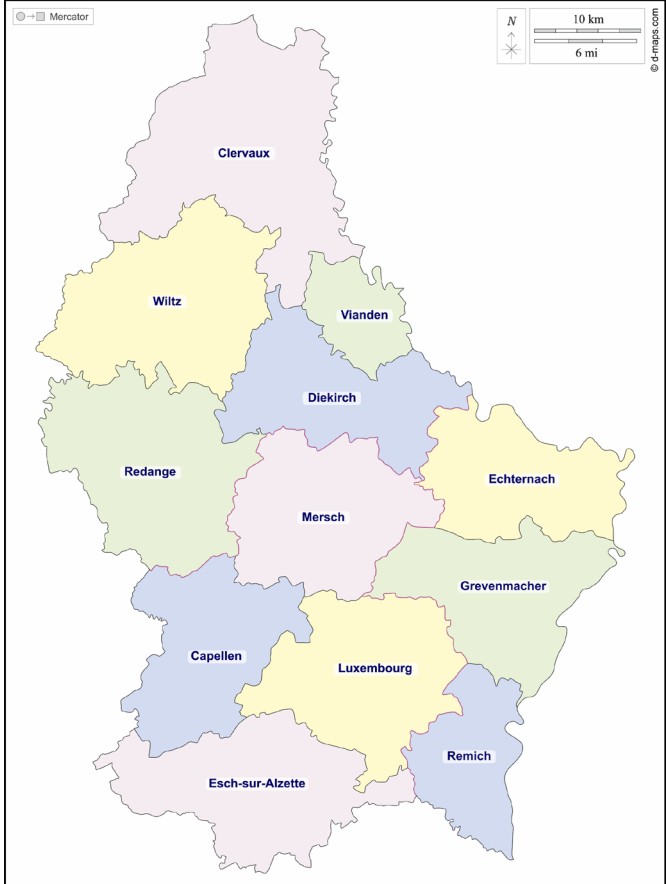

**Figure 1** Map of Luxembourg cantons.

| Table 2 | Prevalence of unmet need | | |
|---|---|---|---|
| **Variable** | **N** | **N, unmet need=1** | **%** |
| Wait | 1639 | 529 | 32 |
| Distance or transportation problems | 1531 | 51 | 4 |
| Could not afford medical care | 1813 | 94 | 5 |
| Could not afford dental care | 1888 | 222 | 12 |
| Could not afford prescribed medicines | 1899 | 108 | 6 |
| Could not afford mental healthcare | 1268 | 63 | 5 |

N refers to number of respondents in estimation sample, N, unmet need=1 refers to the number of respondents in the estimation sample who reported an unmet need for each component and % is the percentage of estimation sample who reported an unmet need for each component weighted for age, sex and district of residence.

We used sample weights to ensure the sample was representative of the population in terms of age, sex and district of residence. We estimated models with robust standard errors. As a sensitivity analysis, following Bataineh *et al*,[50] we included a binary variable equal to one for observations with missing data for income in all models. We reported results for all models as ORs adjusted for the explanatory variables with 95% CI. We estimated all models using Stata V.15.[51]

### Patient and public involvement

Patients or the public were not involved in the design, conduct, reporting or dissemination of the EHIS.

### RESULTS

Table 2 shows the sample size for each component of unmet need and the weighted percentage of unmet need. The most commonly cited barrier to access was due to wait times (32%) while the least common was distance (4%). Fifteen per cent of respondents reported unmet need due to affordability of care. Dental care (12%) was most commonly cited as being unaffordable, followed by prescribed medicines (6%), medical (5%) and mental healthcare (5%).

Table 3 presents the number and weighted percentage of respondents according to the explanatory variables for each component of unmet need. A higher proportion of respondents who had bad or very bad health, a chronic disease, limitations in activities due to health problems or were ex-drinkers reported unmet need for all components. A higher proportion of respondents with obesity, low social support, and whose income fell below the first quintile reported unmet need due to distance and financial barriers. There was no discernible pattern between the remaining respondent characteristics and types of unmet need.

Table 4 displays the adjusted ORs from the multivariate logistic regression models. Respondents who reported bad or very bad health were associated with a higher risk of experiencing unmet need for every component, compared with those with good or very good health. Variables positively associated with reporting unmet need due to wait times included being female (OR 1.52, 95% CI 1.19 to 1.93), having a chronic condition (OR 1.45, 95% CI 1.11 to 1.91) and limitations in activities due to health problems (OR 1.33, 95% CI 1.02 to 1.73). Respondents aged 65 years and over were less likely to experience long wait times in comparison to the 15–24 age group (OR 0.44, 95% CI 0.21 to 0.90). A lower risk of experiencing delays due to distance or transportation problems was associated with the age groups 35–44 (OR 0.20, 95% CI 0.05 to 0.75), 45–54 (OR 0.04, 95% CI 0.01 to 0.23) and 55–64 (OR 0.14, 95% CI 0.03 to 0.63) as well as the income groups between the second and third quintiles (OR 0.19, 95% CI 0.06 to 0.54) and the fourth and fifth quintiles (OR 0.14, 95% CI 0.04 to 0.57). There is some consistency in determinants of unmet need arising from financial barriers for different types of care. Moderate or high social support was associated with a lower risk for dental care, prescribed medicines and mental healthcare. Married respondents were also associated with a lower risk for prescribed medicines (OR 0.53, 95% CI 0.33 to 0.84) and mental healthcare (OR 0.40, 95% CI 0.21 to 0.79). Non-participation in the labour force due to studies, domestic work or compulsory service was associated with a higher risk of being unable to afford prescribed medicines (OR 2.22, 95% CI 1.04 to 4.76) and mental healthcare (OR 2.95, 95% CI 1.28 to 6.77). Income between the first and second quintiles was associated with a lower risk of unmet need for medical care (OR 0.37, 95% CI 0.18 to 0.78) and prescribed medicines (OR 0.49, 95% CI 0.26 to 0.95); income between the third and fourth quintiles was associated with a lower risk for dental care (OR 0.46, 95% CI 0.26 to 0.79) and prescribed medicines (OR 0.33, 95% CI 0.15 to 0.70) and income between the fourth and fifth quintiles was associated with a lower risk for dental care (OR 0.29, 95% CI 0.16 to 0.53), prescribed medicines (OR 0.38, 95% CI 0.17 to 0.82) and mental healthcare (OR 0.17, 95% CI 0.05 to 0.61). Daily smokers were associated with a higher risk of unmet need for dental (OR 1.82, 95% CI 1.28 to 2.60) and mental healthcare (OR 2.15, 95% CI 1.10 to 4.17) compared with non-smokers. Immigrants were at higher risk of being unable to afford prescribed medicines (OR 0.58, 95% CI 0.35 to 0.97).

Online supplemental appendix 1, table A3 shows the results of the sensitivity analysis that included a binary variable measuring income non-response in each model. This variable was statistically significant in only the model for

**Table 3** Characteristics of respondents reporting an unmet need due to wait, distance or affordability

| Variable | Wait (n=1639) | | Distance (n=1531) | | Could not afford medical care (n=1813) | | Could not afford dental care (n=1888) | | Could not afford prescribed medicines (n=1899) | | Could not afford mental healthcare (n=1268) | |
|---|---|---|---|---|---|---|---|---|---|---|---|---|
| | N | % | N | % | N | % | N | % | N | % | N | % |
| **Sex** | | | | | | | | | | | | |
| Male | 215 | 45 | 27 | 57 | 44 | 51 | 106 | 52 | 63 | 61 | 19 | 35 |
| Female | 314 | 55 | 24 | 43 | 50 | 49 | 116 | 48 | 45 | 39 | 44 | 65 |
| **Age** | | | | | | | | | | | | |
| 15–24 | 35 | 8 | 10 | 24 | 6 | 8 | 15 | 8 | 7 | 8 | 4 | 7 |
| 25–34 | 119 | 23 | 11 | 20 | 16 | 17 | 46 | 21 | 21 | 19 | 22 | 36 |
| 35–44 | 119 | 22 | 10 | 18 | 20 | 20 | 46 | 20 | 12 | 11 | 15 | 23 |
| 45–54 | 119 | 21 | 3 | 5 | 25 | 26 | 55 | 25 | 31 | 28 | 10 | 16 |
| 55–64 | 78 | 13 | 6 | 12 | 14 | 14 | 37 | 15 | 16 | 14 | 11 | 17 |
| 65 and over | 59 | 12 | 11 | 21 | 13 | 15 | 23 | 11 | 21 | 20 | 1 | 1 |
| **Marital status** | | | | | | | | | | | | |
| No | 187 | 37 | 24 | 50 | 43 | 46 | 97 | 45 | 51 | 50 | 38 | 62 |
| Yes | 342 | 63 | 27 | 50 | 51 | 54 | 125 | 55 | 57 | 50 | 25 | 38 |
| **Immigrant** | | | | | | | | | | | | |
| No | 322 | 62 | 29 | 56 | 60 | 65 | 134 | 61 | 80 | 73 | 31 | 51 |
| Yes | 207 | 38 | 22 | 44 | 34 | 35 | 88 | 39 | 28 | 27 | 32 | 49 |
| **Education** | | | | | | | | | | | | |
| Primary and pre-primary | 34 | 7 | 5 | 9 | 11 | 12 | 18 | 8 | 12 | 12 | 5 | 7 |
| Secondary and post-secondary | 264 | 50 | 27 | 54 | 66 | 71 | 146 | 66 | 76 | 71 | 37 | 61 |
| Tertiary | 231 | 43 | 19 | 37 | 17 | 17 | 58 | 25 | 20 | 18 | 21 | 32 |
| **Job status** | | | | | | | | | | | | |
| Employed | 334 | 62 | 25 | 47 | 52 | 54 | 136 | 61 | 53 | 47 | 32 | 50 |
| Unemployed | 15 | 3 | 1 | 2 | 8 | 9 | 13 | 6 | 5 | 5 | 7 | 12 |
| Retired | 91 | 17 | 12 | 22 | 16 | 16 | 32 | 14 | 24 | 21 | 4 | 6 |
| Student, domestic, compulsory service | 74 | 15 | 10 | 23 | 11 | 14 | 30 | 14 | 19 | 20 | 13 | 21 |
| Permanently disabled and other inactive status | 15 | 3 | 3 | 6 | 7 | 7 | 11 | 5 | 7 | 7 | 7 | 12 |
| **Social support** | | | | | | | | | | | | |
| Low | 19 | 4 | 4 | 7 | 8 | 8 | 16 | 7 | 10 | 9 | 10 | 16 |
| Moderate | 228 | 44 | 24 | 48 | 39 | 42 | 84 | 39 | 37 | 35 | 35 | 55 |
| High | 282 | 52 | 23 | 45 | 47 | 50 | 122 | 54 | 61 | 56 | 18 | 29 |
| **Household income** | | | | | | | | | | | | |
| Below 1st quintile (lowest) | 100 | 19 | 21 | 40 | 35 | 37 | 67 | 30 | 34 | 32 | 25 | 40 |
| Between 1st and 2nd quintile | 82 | 16 | 13 | 30 | 13 | 15 | 50 | 23 | 18 | 17 | 13 | 20 |
| Between 2nd and 3rd quintile | 133 | 24 | 6 | 11 | 24 | 26 | 57 | 26 | 33 | 30 | 15 | 23 |
| Between 3rd and 4th quintile | 108 | 21 | 7 | 13 | 12 | 13 | 29 | 13 | 12 | 11 | 7 | 12 |
| Between 4th and 5th quintile (highest) | 106 | 20 | 4 | 7 | 10 | 10 | 19 | 8 | 11 | 10 | 3 | 4 |
| **Informal carer** | | | | | | | | | | | | |
| No | 419 | 79 | 43 | 84 | 75 | 80 | 183 | 82 | 88 | 82 | 53 | 84 |
| Yes | 110 | 21 | 8 | 16 | 19 | 20 | 39 | 18 | 20 | 18 | 10 | 16 |
| **BMI** | | | | | | | | | | | | |
| Normal or underweight (<25) | 286 | 53 | 23 | 45 | 47 | 51 | 93 | 42 | 42 | 39 | 28 | 45 |

Continued

**Table 3** Continued

| Variable | Wait (n=1639) | | Distance (n=1531) | | Could not afford medical care (n=1813) | | Could not afford dental care (n=1888) | | Could not afford prescribed medicines (n=1899) | | Could not afford mental healthcare (n=1268) | |
|---|---|---|---|---|---|---|---|---|---|---|---|---|
| | N | % | N | % | N | % | N | % | N | % | N | % |
| Overweight (25–29) | 141 | 28 | 14 | 28 | 19 | 19 | 78 | 34 | 33 | 30 | 19 | 30 |
| Obese (≥30) | 102 | 19 | 14 | 27 | 28 | 30 | 51 | 23 | 33 | 31 | 16 | 25 |
| Smoker | | | | | | | | | | | | |
| No | 422 | 80 | 39 | 76 | 60 | 65 | 150 | 67 | 84 | 78 | 38 | 60 |
| Occasionally | 27 | 5 | 3 | 7 | 10 | 10 | 12 | 5 | 3 | 2 | 2 | 3 |
| Daily | 80 | 15 | 9 | 18 | 24 | 24 | 60 | 28 | 21 | 20 | 23 | 37 |
| Alcohol consumption | | | | | | | | | | | | |
| Never | 102 | 19 | 13 | 25 | 22 | 25 | 55 | 25 | 23 | 22 | 14 | 23 |
| Ex-drinkers | 61 | 11 | 9 | 18 | 15 | 16 | 36 | 17 | 13 | 12 | 12 | 20 |
| Irregularly | 51 | 9 | 9 | 15 | 12 | 12 | 23 | 10 | 16 | 15 | 7 | 10 |
| Regularly | 180 | 34 | 10 | 19 | 27 | 27 | 52 | 24 | 34 | 31 | 19 | 30 |
| Everyday | 135 | 26 | 10 | 23 | 18 | 20 | 56 | 25 | 22 | 20 | 11 | 16 |
| Self-assessed health | | | | | | | | | | | | |
| Good/very good | 325 | 61 | 17 | 34 | 40 | 41 | 113 | 50 | 56 | 51 | 29 | 46 |
| Fair | 142 | 27 | 20 | 40 | 26 | 29 | 74 | 34 | 31 | 30 | 18 | 28 |
| Bad/very bad | 62 | 12 | 14 | 26 | 28 | 30 | 35 | 16 | 21 | 19 | 16 | 26 |
| Chronic disease | | | | | | | | | | | | |
| No | 132 | 25 | 10 | 20 | 25 | 26 | 63 | 28 | 26 | 32 | 8 | 13 |
| Yes | 397 | 75 | 41 | 80 | 69 | 74 | 159 | 72 | 82 | 68 | 55 | 87 |
| Limitations in activities due to health problems | | | | | | | | | | | | |
| No limitations | 272 | 51 | 17 | 35 | 36 | 37 | 106 | 48 | 44 | 40 | 24 | 37 |
| Limited/severely limited | 257 | 49 | 34 | 65 | 58 | 63 | 116 | 52 | 64 | 60 | 39 | 63 |
| Canton | | | | | | | | | | | | |
| Capellen | 54 | 10 | 3 | 6 | 11 | 12 | 24 | 10 | 14 | 12 | 8 | 12 |
| Clervaux | 14 | 3 | 5 | 9 | 4 | 5 | 7 | 4 | 8 | 8 | 2 | 3 |
| Diekirch | 17 | 3 | 5 | 11 | 7 | 8 | 11 | 6 | 5 | 5 | 2 | 3 |
| Echternach | 14 | 3 | 2 | 5 | 4 | 4 | 9 | 4 | 3 | 3 | 2 | 3 |
| Esch sur Alzette | 143 | 26 | 12 | 22 | 30 | 32 | 70 | 32 | 29 | 25 | 18 | 29 |
| Grevenmacher | 37 | 7 | 5 | 10 | 6 | 7 | 15 | 7 | 9 | 9 | 4 | 8 |
| Luxembourg | 171 | 32 | 13 | 26 | 18 | 18 | 55 | 23 | 19 | 17 | 20 | 31 |
| Mersch | 37 | 7 | 1 | 2 | 5 | 5 | 10 | 5 | 9 | 8 | 1 | 1 |
| Redange | 16 | 3 | 1 | 3 | 3 | 3 | 4 | 2 | 2 | 2 | 1 | 1 |
| Remich | 17 | 3 | 2 | 4 | 3 | 3 | 9 | 4 | 5 | 5 | 3 | 6 |
| Vianden | 3 | 1 | 1 | 2 | 1 | 1 | 3 | 1 | 2 | 2 | 1 | 2 |
| Wiltz | 6 | 1 | 1 | 2 | 2 | 3 | 5 | 2 | 3 | 3 | 1 | 1 |

N refers to number of respondents in estimation sample and % is percentage of estimation sample weighted for age, sex and district of residence.
BMI, body mass index.

affordability of dental care with non-reporting of income associated with a lower risk of reporting unmet need due to the affordability of dental care (OR 0.64, 95% CI 0.42 to 0.97). Results for all models were largely unchanged. As in the main analyses, a lower risk of unmet need due to the affordability of dental care was associated with moderate (OR 0.43, 95% CI 0.24 to 0.78) or high (OR 0.40, 95% CI 0.22 to 0.72) social support, income between the third and fourth quintile (OR 0.48, 95% CI 0.28 to 0.81) and fourth

and fifth quintile (OR 0.30, 95% CI 0.17 to 0.54) and regular alcohol consumption (OR 0.60, 95% CI 0.42 to 0.87).

## DISCUSSION
### A statement of the principal findings
In this paper, we investigated the prevalence and determinants of unmet need in Luxembourg using EHIS 2 data. The most common barrier to accessing healthcare

**Table 4** Multivariate logistic regression models for unmet need due to wait, distance or affordability

| Variable | Wait (n=1639) | Distance (n=1531) | Could not afford medical care (n=1813) | Could not afford dental care (n=1888) | Could not afford prescribed medicines (n=1899) | Could not afford mental healthcare (n=1268) |
|---|---|---|---|---|---|---|
| Sex | | | | | | |
| Male | 1 | 1 | 1 | 1 | 1 | 1 |
| Female | 1.52 (1.19 to 1.93)** | 0.83 (0.40 to 1.75) | 1.04 (0.64 to 1.71) | 0.96 (0.69 to 1.33) | 0.55 (0.35 to 0.88)* | 2.89 (1.41 to 5.94)** |
| Age | | | | | | |
| 15–24 | 1 | 1 | 1 | 1 | 1 | 1 |
| 25–34 | 1.32 (0.72 to 2.40) | 0.27 (0.07 to 1.02) | 1.76 (0.49 to 6.27) | 1.62 (0.71 to 3.71) | 3.22 (0.98 to 10.57) | 6.39 (1.75 to 23.35)** |
| 35–44 | 1.04 (0.55 to 1.93) | 0.20 (0.05 to 0.75)* | 1.87 (0.51 to 6.81) | 1.36 (0.58 to 3.16) | 1.53 (0.40 to 5.86) | 2.49 (0.62 to 9.97) |
| 45–54 | 1.00 (0.54 to 1.83) | 0.04 (0.01 to 0.23)*** | 2.09 (0.58 to 7.49) | 1.53 (0.66 to 3.56) | 3.65 (1.05 to 12.66)* | 1.33 (0.31 to 5.60) |
| 55–64 | 0.59 (0.31 to 1.12) | 0.14 (0.03 to 0.63)* | 1.55 (0.39 to 6.06) | 1.51 (0.62 to 3.68) | 2.24 (0.58 to 8.58) | 2.18 (0.51 to 9.35) |
| 65 and over | 0.44 (0.21 to 0.90)* | 0.27 (0.07 to 1.07) | 2.28 (0.50 to 10.36) | 1.21 (0.46 to 3.19) | 2.98 (0.76 to 11.75) | 0.17 (0.01 to 2.16) |
| Marital status | | | | | | |
| No | 1 | 1 | 1 | 1 | 1 | 1 |
| Yes | 1.16 (0.89 to 1.52) | 1.14 (0.47 to 2.78) | 0.73 (0.43 to 1.23) | 0.76 (0.53 to 1.08) | 0.53 (0.33 to 0.84)** | 0.40 (0.21 to 0.79)** |
| Immigrant | | | | | | |
| No | 1 | 1 | 1 | 1 | 1 | 1 |
| Yes | 0.79 (0.61 to 1.01) | 1.18 (0.55 to 2.52) | 0.79 (0.49 to 1.26) | 0.85 (0.61 to 1.19) | 0.58 (0.35 to 0.97)* | 1.37 (0.68 to 2.76) |
| Education | | | | | | |
| Primary and pre-primary | 1 | 1 | 1 | 1 | 1 | 1 |
| Secondary and post-secondary | 1.14 (0.71 to 1.85) | 1.26 (0.42 to 3.79) | 1.48 (0.61 to 3.55) | 1.60 (0.85 to 3.02) | 1.35 (0.64 to 2.85) | 1.55 (0.45 to 5.35) |
| Tertiary | 1.40 (0.82 to 2.37) | 2.59 (0.80 to 8.41) | 0.55 (0.20 to 1.53) | 1.20 (0.59 to 2.44) | 0.67 (0.27 to 1.65) | 1.22 (0.29 to 5.11) |
| Job status | | | | | | |
| Employed | 1 | 1 | 1 | 1 | 1 | 1 |
| Unemployed | 0.62 (0.31 to 1.25) | 0.21 (0.04 to 1.11) | 1.74 (0.59 to 5.11) | 1.4 (0.65 to 3.02) | 1.51 (0.48 to 4.75) | 1.51 (0.45 to 5.09) |
| Retired | 1.22 (0.76 to 1.95) | 0.91 (0.29 to 2.83) | 0.60 (0.23 to 1.54) | 0.58 (0.32 to 1.05) | 0.88 (0.38 to 2.02) | 1.20 (0.30 to 4.71) |
| Student, domestic, compulsory service | 1.25 (0.82 to 1.91) | 0.66 (0.23 to 1.85) | 0.81 (0.33 to 2.01) | 1.01 (0.58 to 1.76) | 2.22 (1.04 to 4.76)* | 2.95 (1.28 to 6.77)* |
| Permanently disabled and other inactive status | 0.80 (0.38 to 1.69) | 1.60 (0.29 to 8.81) | 0.92 (0.35 to 2.41) | 0.90 (0.44 to 1.84) | 1.40 (0.58 to 3.37) | 2.97 (0.92 to 9.64) |
| Social support | | | | | | |
| Low | 1 | 1 | 1 | 1 | 1 | 1 |
| Moderate | 1.54 (0.81 to 2.92) | 0.84 (0.26 to 2.72) | 0.60 (0.22 to 1.66) | 0.46 (0.23 to 0.95)* | 0.32 (0.14 to 0.77)* | 0.27 (0.09 to 0.76)* |
| High | 0.95 (0.50 to 1.81) | 0.64 (0.19 to 2.22) | 0.44 (0.16 to 1.23) | 0.42 (0.21 to 0.86)* | 0.30 (0.13 to 0.70)** | 0.11 (0.03 to 0.35)*** |
| Household income | | | | | | |

Continued

**Table 4** Continued

| Variable | Wait (n=1639) | Distance (n=1531) | Could not afford medical care (n=1813) | Could not afford dental care (n=1888) | Could not afford prescribed medicines (n=1899) | Could not afford mental healthcare (n=1268) |
|---|---|---|---|---|---|---|
| Below 1st quintile (lowest) | 1 | 1 | 1 | 1 | 1 | 1 |
| Between 1st and 2nd quintile | 0.85 (0.57 to 1.26) | 0.74 (0.31 to 1.76) | 0.37 (0.18 to 0.78)** | 0.77 (0.48 to 1.21) | 0.49 (0.26 to 0.95)* | 0.49 (0.22 to 1.12) |
| Between 2nd and 3rd quintile | 1.01 (0.70 to 1.48) | 0.19 (0.06 to 0.54)** | 0.73 (0.40 to 1.36) | 0.77 (0.49 to 1.21) | 0.77 (0.43 to 1.36) | 0.78 (0.35 to 1.72) |
| Between 3rd and 4th quintile | 1.16 (0.77 to 1.73) | 0.35 (0.11 to 1.10) | 0.47 (0.22 to 1.01) | 0.46 (0.26 to 0.79)** | 0.33 (0.15 to 0.70)** | 0.47 (0.17 to 1.29) |
| Between 4th and 5th quintile (highest) | 0.91 (0.59 to 1.38) | 0.14 (0.04 to 0.57)** | 0.46 (0.20 to 1.07) | 0.29 (0.16 to 0.53)*** | 0.38 (0.17 to 0.82)* | 0.17 (0.05 to 0.61)** |
| Informal carer | | | | | | |
| No | 1 | 1 | 1 | 1 | 1 | 1 |
| Yes | 1.36 (1.02 to 1.82)* | 0.92 (0.42 to 2.03) | 1.17 (0.67 to 2.03) | 0.96 (0.63 to 1.45) | 0.91 (0.53 to 1.56) | 0.72 (0.32 to 1.64) |
| BMI | | | | | | |
| Underweight or normal | 1 | 1 | 1 | 1 | 1 | 1 |
| Overweight | 0.81 (0.62 to 1.06) | 1.04 (0.49 to 2.22) | 0.49 (0.27 to 0.90)* | 1.28 (0.90 to 1.83) | 1.17 (0.69 to 1.99) | 1.92 (0.94 to 3.91) |
| Obese | 0.99 (0.72 to 1.35) | 1.24 (0.61 to 2.55) | 1.09 (0.62 to 1.92) | 1.35 (0.89 to 2.02) | 1.72 (0.96 to 3.08) | 1.49 (0.68 to 3.27) |
| Smoker (no) | | | | | | |
| No | 1 | 1 | 1 | 1 | 1 | 1 |
| Occasionally | 1.06 (0.64 to 1.74) | 1.38 (0.43 to 4.43) | 2.76 (1.27 to 5.99)* | 0.96 (0.49 to 1.86) | 0.39 (0.10 to 1.44) | 0.35 (0.07 to 1.64) |
| Daily | 0.91 (0.66 to 1.26) | 0.87 (0.38 to 1.96) | 1.25 (0.72 to 2.15) | 1.82 (1.28 to 2.60)** | 0.90 (0.54 to 1.50) | 2.15 (1.10 to 4.17)* |
| Alcohol consumption | | | | | | |
| Never | 1 | 1 | 1 | 1 | 1 | 1 |
| Ex-drinkers to drinkers | 1.49 (0.95 to 2.35) | 1.09 (0.41 to 2.90) | 1.33 (0.59 to 3.01) | 1.60 (0.93 to 2.77) | 1.10 (0.49 to 2.49) | 1.92 (0.73 to 5.08) |
| Irregularly | 1.06 (0.68 to 1.63) | 1.68 (0.64 to 4.38) | 1.22 (0.54 to 2.73) | 0.89 (0.51 to 1.55) | 1.19 (0.59 to 2.41) | 1.27 (0.40 to 4.04) |
| Regularly | 1.12 (0.81 to 1.55) | 0.75 (0.29 to 1.94) | 0.89 (0.48 to 1.65) | 0.63 (0.41 to 0.97)* | 0.97 (0.54 to 1.74) | 1.36 (0.59 to 3.17) |
| Everyday | 1.07 (0.75 to 1.53) | 0.97 (0.37 to 2.53) | 0.85 (0.41 to 1.74) | 1.02 (0.65 to 1.59) | 0.80 (0.41 to 1.56) | 1.20 (0.49 to 2.98) |
| Self to assessed health | | | | | | |
| Good/very good | 1 | 1 | 1 | 1 | 1 | 1 |
| Fair | 1.21 (0.90 to 1.62) | 2.45 (1.17 to 5.12)* | 1.26 (0.66 to 2.43) | 1.63 (1.10 to 2.40)* | 0.9 (0.52 to 1.55) | 1.01 (0.44 to 2.28) |
| Bad/very bad | 2.41 (1.53 to 3.80)*** | 7.12 (2.91 to 17.44)*** | 5.35 (2.39 to 11.95)*** | 3.26 (1.86 to 5.71)*** | 2.22 (1.04 to 4.71)* | 3.58 (1.25 to 10.30)* |
| Chronic disease | | | | | | |
| No | 1 | 1 | 1 | 1 | 1 | 1 |
| Yes | 1.45 (1.11 to 1.91)** | 1.42 (0.60 to 3.37) | 0.69 (0.38 to 1.28) | 0.83 (0.57 to 1.21) | 0.84 (0.48 to 1.44) | 2.33 (0.89 to 6.08) |
| Limitations in activities due to health problems | | | | | | |

Continued

**Table 4** Continued

| Variable | Wait (n=1639) | Distance (n=1531) | Could not afford medical care (n=1813) | Could not afford dental care (n=1888) | Could not afford prescribed medicines (n=1899) | Could not afford mental healthcare (n=1268) |
|---|---|---|---|---|---|---|
| No limitations | 1 | 1 | 1 | 1 | 1 | 1 |
| Limited/severely limited | 1.33 (1.02 to 1.73)* | 1.37 (0.67 to 2.81) | 1.39 (0.74 to 2.64) | 1.15 (0.79 to 1.67) | 1.49 (0.86 to 2.58) | 1.53 (0.64 to 3.66) |
| Canton | | | | | | |
| Esch zur Alzette | 1 | 1 | 1 | 1 | 1 | 1 |
| Capellen | 1.21 (0.80 to 1.82) | 1.08 (0.25 to 4.57) | 1.42 (0.64 to 3.13) | 1.41 (0.83 to 2.40) | 2.12 (1.04 to 4.30)* | 1.63 (0.61 to 4.37) |
| Clervaux | 0.90 (0.46 to 1.75) | 3.31 (0.78 to 14.14) | 1.08 (0.31 to 3.81) | 0.76 (0.31 to 1.88) | 2.06 (0.77 to 5.53) | 0.53 (0.12 to 2.31) |
| Diekirch | 0.92 (0.49 to 1.72) | 3.54 (1.02 to 12.32) | 2.27 (0.87 to 5.95) | 1.31 (0.60 to 2.83) | 1.88 (0.68 to 5.18) | 0.67 (0.12 to 3.85) |
| Echternach | 1.43 (0.69 to 2.97) | 2.02 (0.36 to 11.36) | 1.63 (0.40 to 6.67) | 1.47 (0.68 to 3.17) | 1.13 (0.29 to 4.37) | 1.19 (0.14 to 10.34) |
| Grevenmacher | 1.67 (1.01 to 2.78) | 3.06 (1.02 to 9.15) | 1.38 (0.49 to 3.90) | 1.27 (0.65 to 2.49) | 2.53 (1.08 to 5.96)* | 1.28 (0.30 to 5.51) |
| Luxembourg | 1.27 (0.94 to 1.72) | 1.40 (0.59 to 3.30) | 0.80 (0.42 to 1.50) | 0.99 (0.65 to 1.50) | 1.02 (0.54 to 1.94) | 1.54 (0.73 to 3.25) |
| Mersch | 1.67 (1.01 to 2.76)* | 0.43 (0.05 to 3.74) | 1.11 (0.39 to 3.22) | 0.88 (0.40 to 1.94) | 2.38 (1.00 to 5.67) | 0.29 (0.04 to 2.23) |
| Redange | 1.07 (0.56 to 2.04) | 0.95 (0.09 to 10.02) | 0.91 (0.27 to 3.04) | 0.53 (0.17 to 1.60) | 0.60 (0.13 to 2.80) | 0.80 (0.08 to 7.68) |
| Remich | 0.99 (0.52 to 1.90) | 1.64 (0.32 to 8.40) | 0.75 (0.23 to 2.40) | 1.05 (0.50 to 2.20) | 1.51 (0.58 to 3.97) | 3.45 (0.94 to 12.74) |
| Vianden | 0.59 (0.14 to 2.46) | 6.92 (0.62 to 77.07) | 0.80 (0.14 to 4.45) | 1.04 (0.29 to 3.67) | 2.13 (0.58 to 7.77) | 1.64 (0.24 to 11.19) |
| Wiltz | 0.39 (0.15 to 0.99) | 0.45 (0.07 to 3.16) | 1.79 (0.33 to 9.75) | 0.95 (0.34 to 2.71) | 2.35 (0.57 to 9.65) | 0.40 (0.05 to 3.14) |
| Constant | 0.15 (0.05 to 0.42)*** | 0.08 (0.01 to 0.67)* | 0.10 (0.02 to 0.57)* | 0.22 (0.06 to 0.78)* | 0.14 (0.03 to 0.59)** | 0.01 (0.00 to 0.11)*** |

Estimates are weighted for age, sex and district of residence. Data are OR adjusted for explanatory variables (95%CI).
*P<0.05, **p<0.01, ***p<0.001.
BMI, body mass index.

arose from long waits (32%). Compared with other EU countries who collected this data using the EHIS, Luxembourg had the highest proportion of respondents who reported unmet need due to long waits.[52] Respondents were most likely to report being unable to afford dental care (12%), followed by prescribed medicines (6%)—a result that is on par with the EU averages (12.3% and 4.6%, respectively).[53] The percentage of respondents who reported being unable to afford mental healthcare (5%) was higher than the EU average (2.7%).[53] The association of various determinants of unmet need varied according to the different components. However, bad or very bad self-reported health was positively associated with every component of unmet need. This group was twice as likely to report unmet need due to wait time, seven times more likely to report unmet need due to distance or transportation, and two to five times more likely to report unmet need due to affordability of services compared with respondents who assessed their health as good or very good. Another notable finding was the clear income gradient for unmet need due to the affordability of care, with the highest income quintile 71% less likely to report unmet need for dental care, 62% less likely for prescribed medications and 83% less likely for mental healthcare compared with the lowest income quintile.

## Comparisons with previous studies

Previous studies[14 16 21 23 25] have investigated unmet need within countries using EU-SILC data. The use of the EHIS brought two important advantages that moved us beyond these studies. First, we undertook separate analyses of unmet need arising from long waits, distance or transportation problems, and affordability of medical, dental and mental healthcare and prescribed medicines. Second, we exploited a richer set of health variables, which enabled us to investigate risk factors including BMI, smoking and alcohol consumption as determinants of unmet need. Moreover, this is the first study that used the EHIS to investigate the determinants of unmet need in the general population. Previous studies used the EHIS to investigate unmet need for specific population groups. Sakellariou et al[30] investigated whether people with disabilities had higher unmet need for healthcare due to waiting times, distance or transport problems and unaffordability of medical and mental healthcare and prescribed medicines compared with people without disabilities in the UK. Rotarou et al[29] conducted a similar study for Greece. Hoebel et al[28] focused only on respondents aged 50–85 years of age in Germany to investigate unmet need due to the affordability of medical, dental and mental healthcare and prescribed medicines. They did not consider long waits or distance or transportation problems.

Three previous studies[14 19 23] used EU-SILC data on reasons for unmet need to investigate associations with various determinants. Two studies[14 23] considered the main reason for unmet need due to availability (waiting lists) and accessibility (affordability and distance/transportation) while the remaining study[19] considered

affordability and distance/transportation as separate categories. Comparable to our results, these studies reported a positive association between bad and very bad health (compared with very good) and unmet need due to availability[14] and accessibility[14 23] or only affordability of care.[19] One study[19] found that social support as measured by the ability to seek help was associated with a lower probability of unmet needs for medical care arising from economic costs, reflecting our finding that higher levels of social support were associated with lower unmet need due to affordability of care. Our result that females were associated with higher unmet need due to wait times was not supported by these studies, which found evidence of higher unmet need for females due to accessibility[14] and affordability of care.[19] Women and respondents with bad or very bad self-assessed health may have been unable to afford mental healthcare because they have higher need. A previous study[49] investigated the burden of depression in Luxembourg and reported a higher prevalence rate of depression symptoms in females compared with males and in those who perceived their health as poor, compared with those who perceived their health as good. This study also found that good social support was associated with a lower risk of depression, which could indicate a lower need for mental healthcare and hence lower unmet need arising from the affordability of mental healthcare.

Similar to previous studies,[14 23] we found that higher income was associated with a lower risk of unmet need due to distance and affordability of services. Income disparities in unmet need due to the affordability of services may have reflected the requirement for patients to pay the full costs of many services directly to providers at the point of use. Our finding of no statistically significant relationship between income and unmet need due to waits was also reflected in previous studies.[14 16 19 23] Although we did not find any association between unmet need due to waits and socioeconomic variables including education, income and job status, previous studies reported evidence of inequalities in waiting times related to education and income.[54–56] The EHIS data allowed us to consider the association between health behaviours and unmet need. We found that these variables were only associated with unmet need arising from the affordability of healthcare. Therefore, our results add to a previous study from Canada[50] that considered the relationship between obesity, drinking and smoking and unmet need due to health system factors (including unavailability of services and long waiting times and excluding cost) and found evidence of a positive association between smoking (both daily and occasional) and unmet need.

## Study limitations

Our study is subject to limitations arising from our data, some of which could in principle be overcome in future research, via development and adjustment of the EHIS questionnaire. As unmet need was self-reported, the data may suffer from the limitations inherent in survey data, including recall or response bias (respondent inaccurately

remembers or misunderstands the question).[57 58] Moreover, self-reported unmet need may be influenced by unobservable factors such as cultural norms, health literacy and expectations of health services.[15 20 59 60] The EHIS did not ask respondents if they perceived long waits across the healthcare system or for particular services (primary care, cancer care, elective treatment, diagnostic tests), nor the length of time perceived as long. The survey sampling did not include sections of society who lack any insurance coverage and may have high unmet need including the homeless and irregular immigrants and their families.[34] Individuals living in collective households (eg, a boarding house or hostel or a dormitory in an educational establishment)[43] and institutions were also excluded. Despite these limitations, survey data remains the best available source.[59] We excluded observations with missing data for household income. The profile of respondents who did not report household income indicated that they were young people in full-time education who may have been living in the family home. Perhaps these respondents did not know the value of the household income, because they were not a main earner. We could assume that some of these respondents were less likely to bear the financial costs of care and therefore less likely to report unmet need due to the affordability of care. Indeed, the results of the sensitivity analysis as shown in online supplemental appendix 1, table A3 suggested this was the case, although the variable measuring income non-response is only statistically significant in the model measuring affordability of dental care.

### Implications for clinicians and policy-makers

The Luxembourgish health system is characterised by a high rate of population health insurance coverage, a comprehensive benefit package and a relatively low level of out-of-pocket payments. Nevertheless, access to healthcare is an important issue, particularly in relation to wait times, as reflected in our finding that the majority of respondents reported unmet need due to long waiting times to obtain an appointment. While Luxembourg has introduced reforms to address waiting times in emergency departments and for cancer care, there is potential to implement measures that could address waiting times for other priority services where patients must wait to obtain an appointment, for example, primary care. Unlike Luxembourg, in many European countries, nurses or physician assistants play a key role in primary care by providing health education, immunisations and routine checks of people with chronic illnesses,[8] allowing general practitioners more time to attend to patients with more complex needs. Policy-makers could also consider the introduction of a maximum waiting time for specialist consultations, a policy adopted by several European countries.

In response to the COVID-19 pandemic, teleconsultations are now reimbursed by the CNS. The retention and embedding of telemedicine into the healthcare system in Luxembourg could help to reduce unmet need due to distance and transportation, particularly for those with poor health who may face difficulties in travelling to healthcare providers. The increased use of teleconsultations could also help to address waiting times for particular services, for example, primary and specialist care.

While the introduction of a universal third-party payment system may help to address unmet need due to the affordability of care this reform may not be sufficient by itself as individuals may still face a financial barrier to care due to copayments. For example, while prescribed medicines are currently covered by the third-party payment system, meaning that individuals only pay the copayment, our results show that 6% of respondents reported unmet need to affordability of prescribed medicines. Therefore, additional reforms aimed at reducing the risk of unmet need could address copayments and target high-risk groups such as those with low incomes or in poor health, similar to policies in Belgium and France (as described in the Introduction section). Currently the CNS does not reimburse visits to psychologists and expanding health insurance coverage to psychologists could help to address unmet need due to the affordability of mental healthcare, especially for at-risk groups including individuals who are female, aged 25–34, students or undertaking domestic duties or compulsory service and with low income.

### Implications for future research

Future revisions of the EHIS could expand the social variables collected, for example, by adding information on ethnicity or parents' country of birth. Additional questions on social support could help to distinguish between emotional, instrumental and informational support.[19] Future research could investigate the issue of waiting times in Luxembourg for different types of services in order to inform policy measures to reduce unmet need. Recent policies focused on the reorganisation of cancer care delivery and introduction of waiting time targets for cancer care could be evaluated. Given our finding of income disparities in unmet need due to the affordability of care, an evaluation of the implementation of the third-party payment system to date would inform its planned expansion and potential further adjustments. The collection of a third wave of EHIS data will enable an assessment of trends in unmet need in Luxembourg over time. This research also demonstrates that the EHIS is a useful resource that could be used for future within-country and between-country analyses of unmet need for healthcare in the general population in European countries. While previous cross-country studies have examined patient and health system characteristics associated with unmet need across European countries for medical and dental care, the EHIS could complement these studies by extending the analyses to unmet need arising from the affordability of mental healthcare and prescribed medicines.

## CONCLUSIONS

Compared with other EU countries, Luxembourg has high per capita public healthcare expenditures, a low share of total health spending financed by out-of-pocket payments and a comprehensive benefit package.[61][62] Nevertheless, this study provides evidence that some individuals experienced difficulties accessing care, particularly due to long waits and affordability of care. Future reforms to improve access to healthcare should first target high-risk groups including those with low incomes and poor health. Future policies to address unmet need should also consider other vulnerable populations who lack formal healthcare coverage including the homeless, undocumented immigrants and those in the informal labour market.

**Author affiliations**
[1]Department of Population Health, Luxembourg Institute of Health, Strassen, Luxembourg
[2]Living Conditions department, Luxembourg Institute of Socio-Economic Research, Esch-sur-Alzette/Belval, Luxembourg
[3]Centre for Health Economics, University of York, York, UK
[4]Health and Health Systems, Luxembourg Institute of Socio-Economic Research, Esch-sur-Alzette/Belval, Luxembourg
[5]Service Nomenclature, conventions, analyse et prospective, Caisse nationale de santé, Luxembourg, Luxembourg
[6]Direction générale, Santé publique France, Saint-Maurice, France

**Acknowledgements** We would like to thank to the population of Luxembourg and all the EHIS team who have contributed to this study. We would like to thank EHIS steering committee members S Couffignal, N de Rekeneire, S Leite, G Weber, AC Lorcy, G Osier and I Salagean for their valuable contributions. We thank colleagues for their feedback on earlier iterations of the research presented at seminars at the Luxembourg Institute of Health and Luxembourg Institute of Socio-Economic Research. We also thank the reviewers for their valuable comments.

**Contributors** VM: conceptualisation, methodology, software, formal analysis, writing-original draft, writing-review and editing, visualisation. MS: writing-review and editing, MR-C: writing-review and editing, software, resources. JB: data curation, LH: conceptualisation, writing-review and editing, supervision.

**Funding** The EHIS Luxembourg study was funded by the Luxembourg Ministry of Health Directorate of Health and the Luxembourg Institute of Health. This research was supported by the Luxembourg National Research Fund (C19/BM/13723812).

**Disclaimer** The views presented are those of the authors alone and not of the CNS.

**Map disclaimer** The inclusion of any map (including the depiction of any boundaries therein), or of any geographic or locational reference, does not imply the expression of any opinion whatsoever on the part of BMJ concerning the legal status of any country, territory, jurisdiction or area or of its authorities. Any such expression remains solely that of the relevant source and is not endorsed by BMJ. Maps are provided without any warranty of any kind, either express or implied.

**Competing interests** JB was formerly an employee of the Luxembourg Institute of Health and is currently employed at the CNS.

**Patient and public involvement statement** Patients or the public were not involved in the design, conduct, reporting or dissemination of the EHIS.

**Patient consent for publication** Not required.

**Ethics approval** Luxembourg's National Ethics Committee, the Comité National d'Ethique de Recherche (CNER) was informed by the Luxembourg Ministry of Health Directorate of Health of the national regulation and European obligation to collect EHIS data. The survey was completely anonymous and no information on name, address or date of birth of respondents was collected.

**Provenance and peer review** Not commissioned; externally peer reviewed.

**Data availability statement** Data may be obtained from a third party and are not publicly available. Information on the data access procedure for the 2014 EHIS for Luxembourg is available on the Ministry of Health website: https://sante.public.lu/fr/statistiques/ehis/ehis-methodologie/ehis-formulaire-demande-de-donnees.docx.

**ORCID iD**
Valerie Moran http://orcid.org/0000-0002-7168-2059

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
