## [Reviewer comments · BMJ Open]

ARTICLE DETAILS

TITLE (PROVISIONAL)	Investigating unmet need for health care using the European Health Interview Survey: a cross-sectional survey study of Luxembourg
AUTHORS	Moran, Valerie; Suhrcke, Marc; Ruiz-Castell, Maria; Barré, Jessica; Huiart, Laetitia

VERSION 1 – REVIEW

REVIEWER	Francisco Reyes-Santías Universidad de Vigo, Organización de Empresas e Mercadotecnia
REVIEW RETURNED	27-Jan-2021

GENERAL COMMENTS	The paper "Investigating unmet need for health care using the European Health Interview Survey: a cross-sectional survey study of Luxembourg", studies the determinants of the unmet demand for healthcare through the information contained in the European Health Interview Survey (EHIS) for Luxembourg. Title and summary. The title and abstract express well the object of study, objectives and results of the article. Structure of the article. The contents are well organized and they adhere to the IMRaD structure. It include a theoretical framework of the research problem but at this point I suggest the authors incorporate a bibliographic reference that I miss in the text: Hintzpeter B, Finger JD, Allen J (2019) European Health Interview Survey (EHIS) 2 – Background and study methodology. Journal of Health Monitoring 4(4):66-79. Focusing the opportunity of the study, it must be said that it is a useful work sBecause the paper studies the possible determinants of difficulties in accessibility and affordability, taking into account sociodemographic characteristics of the population under study, lifestyle habits and characteristics of the health system such as copayments or waiting lists.. In that sense, hypotheses could be established in a clearer way for the reader than as they are intended to be established (i.e. "unmet needs vary according to services" -what type of services: surgical, hospitalization, or primary care, etc. -) (page 4, lines 47-52). Materials and methods. Regarding the material and methods section, the methodology is tailored to the object of study and the objectives and is explained in a transparent manner while it has been validly applied to guarantee the results. However, I suggest that the authors clarify some issues in the text: 1.- It would be interesting if authors indicated that the 4000 subjects in the sample refer to people who have not shown a need for health care and those who have. otherwise, the exclusion of people who have not required healthcare would lead to higher ratios of unmet needs for the rest of the population.2.- Authors could also indicate for readers unfamiliar with the EHIS
--

	questionnaire that, unlike other questionnaires such as the EU-SILC or the Commonwealth Fund International Health Policy Survey, it directly asks all respondents about the barriers they have encountered in receiving care. not if they needed such care in the first place. 3.- Authors are suggested to incorporate the expression of the multivariate logistic regression function that they use to analyze unmet needs due to waiting times, distance and affordability. Results. The results are significant and they are presented in an adequate and understandable way not only through narration, but also with self-explained tables and figures that are also well elaborated in terms of presentation. The results justify and relate to the objectives and methods and the results are of sufficient social interest. Discussion. The discussion appropriately compares the study results with other works, highlighting the main study findings. 75% of the bibliography cited in the study belongs to the previous five years. But it is suggested to the authors that the discussion incorporates the topic of the impact of socioeconomic variables on the waiting time to access health care; such as the study by Simonsen et al: Simonsen NF, Oxholm AS, Kristensen SR, Siciliani L. What explains differences in waiting times for health care across socioeconomic status? Health Econ. 2020 Dec;29(12):1764-1785. doi: 10.1002/hec.4163. Epub 2020 Sep 29. PMID: 32996212. The conclusions are adequately related to the objective. Overall, it is an interesting study, and should be considered for publication in BMJ Open, once the minor revisions proposed have been resolved.
--	--

REVIEWER	Justinem De Oliveira Duke University, Family Medicine and Community Health
REVIEW RETURNED	28-Jan-2021

GENERAL COMMENTS	Well described research strategy, well written manuscript, thoughtful discussion of findings. A useful framework for comparison of access to care across EU countries. Social indicators could be expanded in subsequent revisions to EHIS. While focused on the US, the Andersen model (Aday LA, Andersen R. A framework for the study of access to medical care. Health Serv Res. 1974;9(3):208–220.) and its subsequent revisions may offer food for thought, and expanding consideration of predisposing factors and patient satisfaction and their impact on access to care.
--

REVIEWER	Lucina Rolewicz Nuffield Trust
REVIEW RETURNED	05-Mar-2021

GENERAL COMMENTS	In general this is a well-written paper, and I think it will add to the landscape of literature in this area. Abstract Objectives – this is slightly unclear. Is the aim of the paper to “illustrate the value of the EHIS” or rather is it to investigate the barriers most associated with patient-reported unmet need in healthcare, and the relationship between patient characteristics and the components of unmet need? If the former, then the results should reflect how “value of the EHIS” is measured.
---

	It could also be more concise – can mention that this focuses on Luxembourgish patients (or patients residing in Luxembourg) in a preceding sentence and exclude the last part of line 7 (page 3) “...as an interesting and policy-relevant case study.” Design – statistical methods are not relevant to the study design and can be omitted. Settings and participants – are the participants registered with health care services in Luxembourg? This could be clarified in the text. Conclusions – again, this doesn’t seem to be the main aim of the paper – I would say the breadth of the unmet need definition is just a strength of the EHIS survey. The results are showing which patient group (i.e. those with poor health) is associated with the highest levels of unmet need. It is not clear how the usefulness of the EHIS in measuring unmet need is a novel finding, as it has been used previously, as outlined on page 5, line 25. I understand that previous studies have used the survey to analyse specific sub-groups, but generally using the EHIS as a measure of unmet need has been done before. The conclusion should set out how the main results can help address unmet need amongst patients in Luxembourg. Introduction In general, there is a lot of unnecessary detail in the introduction. The introduction could benefit from a clearer structure on, for example, why is measuring unmet need important; the literature demonstrating unmet need in healthcare and the implications unmet need has on patients/the system, specifically in Luxembourg; where unmet need arises from (i.e. wait times, affordability, distance/transportation, any other factors beyond the scope of the survey?), noting any variation between patient groups and healthcare services; and specific ways that government and/or health services are trying to address the different aspects of unmet need. This should touch on areas such as policies aimed at improving waiting times (which are mentioned in the discussion section), seeing as it is a medium-high priority in Luxembourg and is an indicator of unmet need, and distance/transportation (if policies exist e.g. digital transformation of care). Page 4, lines 22-52 could be written more concisely. Sentences on lines 34-42 could be reworded and possibly removed. This paragraph would also sit better at the end of the introduction section as it will flow better into the study methods. Page 4, line 45 “potentially more thorough way” sounds too doubtful, would consider removing “potentially”. Page 4, lines 47-52 – it isn’t entirely clear why Luxembourg is interesting relative to other countries. The authors mention different cost barriers, but it isn’t clear what these distinct costs are. Page 5, lines 3-45 – this level of detail about funding is not necessary, given the stated aim of the paper. Some background is useful, but the important point here is how funding mechanisms described relate to unmet need in patients (page 6 line 47 – page 7 line 13). Belgium and France are mentioned as examples where some population groups have their costs covered. It is more important to explicitly mention if this is the case in Luxembourg – or is the cost of healthcare the same for all patients? The authors have mentioned in the next paragraph that those in economic hardship can request assistance with payments, but this would benefit from being mentioned in the preceding paragraph as a comparison with Belgium and France, and how this helps (if at all) address unmet need. Also, it is worth clarifying whether the universal third-party payment system replaces the third-party social payment. Data and methods Page 6, line 53 – it’s not entirely clear from the text if 4,004
--	--

	individuals reported a need for care in the survey. If not, another value representing the number of respondents indicating a need for care would be useful here. Page 7, lines 28-38 – the authors should mention whether these models are unadjusted or adjusted and do they include fixed effects or random effects? Results Page 8, lines 22-32 – should the wording here be “A higher proportion of respondents who had bad or very bad health, a chronic disease, limitations in activities due to health problems or were ex-drinkers reported unmet need for all components” as it relates to frequencies and percentages? The same applies for the following sentence in the paragraph. Page 9, lines 30-35 – which group were less likely to experience unmet need with respect to affordability of dental care? This should be explicitly stated in the text. Discussion It is unclear how comparisons with European countries (not previously mentioned in the paper) relate to discussing the main findings of the study. Since the introduction drew on comparisons between Luxembourg and France/Belgium, it may be better (if possible) to continue with this comparison into the discussion. Some reorganisation of this section may benefit the reader. For example, starting with reiterating the main results; followed by comparisons with previous studies and strengths and limitations of this study; followed by implications for policy and clinical practice; and finishing with implications for future research. Could the results of this paper be used to target patients with specific characteristics that are most at-risk of unmet need? Page 10, line 12 – the part of the sentence in brackets doesn't seem to make sense – would consider revising. Page 10, line 15 – health is spelled incorrectly. Page 10, line 20 – the authors mention that individuals living in collective households were excluded. This could be my misunderstanding, but would this exclude young people (e.g. aged 15-18) who live in the family home? If the study includes these patients, is the finding that a higher proportion of the younger age group have missing income data an expected finding, since they are less likely to have an income and are also less likely to bear the financial cost of care? If either of these points are true, this would be worth clarifying in the text. Page 10 – one limitation of the results is that they cannot compare unmet need arising from waiting times across different types of healthcare services. Waiting times from different services are mentioned in detail in the discussion so the authors should highlight the limitations in how far the study data is able to explain these differences. Page 12, lines 19-20 – I am not sure how helpful it is to compare waiting targets between Luxembourg and England – I would imagine the population needs are very different, as one example. Other comments Table 1 – does the percentage refer to the weighted percentage of patients who reported unmet need for each individual component? It may be worth clarifying this in the note. The authors should try and write the paper in past tense. Most of the paper is written in the present tense.
--	---

VERSION 1 – AUTHOR RESPONSE

Response to reviewers

We would like to thank the editor and the three reviewers for taking the time to read the article and provide very valuable feedback, which we feel has strengthened the paper. We have incorporated the editor and reviewers' suggestions and hope they are satisfactory. As there is some overlap in comments, we have organised our response according to the abstract and each section of the paper. We also include the full list of editor and reviewers' comments at the end of this document with the section and page number where the comment is addressed in the revised paper.

Abstract

The Editor and Reviewer 3 asked us to revise the Abstract in order to clarify the objectives, omit statistical methods from the design, clarify that participants were registered with health care services in Luxembourg and revise the Conclusions to reflect the findings on unmet need for health care in Luxembourg.

We have revised the Abstract as recommended. We have completely rewritten the Objectives section to clarify that our aim was to investigate the prevalence of unmet need arising from wait times, distance/ transportation and financial affordability using the European Health Interview Survey and explore associations between individual characteristics and the probability of reporting unmet need. We added our findings on income to the Results section and focused the Conclusions section on the implications of our results for policymakers in Luxembourg.

Introduction

We have extensively revised the Introduction in response to the comments received.

Reviewer 3 noted that the introduction could benefit from a clearer structure and removal of unnecessary detail.

We have revised and restructured the text as recommended by Reviewer 3. We have included sub-headings reflecting the new structure: '*Previous literature on unmet need for health care*' and '*The Luxembourg health system and access to health care*'. We have added additional text on:

- the European Pillar of Social Rights to explain why measuring unmet need is important (page 4),
- unmet need arising from factors other than waiting times, distance/transportation and affordability of care (page 4),
- policies to address unmet need due to waiting times and financial affordability, focusing on the neighbouring countries of Belgium and France, which have similar health systems to Luxembourg (page 4),
- a summary of the literature on unmet need focusing on the individual characteristics associated with unmet need (pages 4-5).

On pages 5-7 under the subheading '*The Luxembourg health system and access to health care*', we have deleted and reorganized some of the previous text on funding. We have also revised the text explaining how current funding mechanisms relate to unmet need. We have moved text on the issue of waiting times in Luxembourg from the Discussion section to the Introduction section. We feel that the revised text presents an appropriate context and helps to make a stronger argument for why it is of relevance and interest to explore the issue of unmet need in Luxembourg and the motivation for our study.

The Editor and Reviewer 1 asked us to state clearly our objectives, hypotheses and research question at the end of the Introduction section and we have added the following text on page 7:

“The objective of this paper is to investigate the prevalence and determinants of unmet need in Luxembourg. We use EHIS data as it allows us to explore the prevalence and determinants of unmet need separately for waiting time, distance or transportation and the affordability of medical, dental and mental health care and prescribed medicines. Therefore, we also contribute to the limited number of studies that investigated unmet need not only for medical care but also for dental and mental healthcare and prescribed medications.”

Data and Methods

We have included the following sub-headings in this section to guide the reader: ‘*Study population and design*’, ‘*Outcome variables*’, ‘*Explanatory variables*’ and ‘*Statistical data analysis*’.

The Editor asked for clarification on ‘participation rate’ and ‘response rate’. We have clarified on page 8 of the revised manuscript that the response rate is 30.1% and the final participation rate is 25%:

“Among the 16,000 individuals invited to participate, 4,823 responded (response rate of 30.1%) by submitting an electronic (70%) or paper (30%) questionnaire [46]. Of these respondents, 4,118 participants met the inclusion criteria, provided informed consent and completed the questionnaire (participation rate of 24.7%). Data were collected between February and December 2014 [47]. The EHIS 2 Luxembourg database comprised 4,004 individuals who completed more than 50% of the questionnaire and had no missing data for age, sex or district, (final participation rate of 25%). This database was prepared according to a European protocol [48] and was validated by Eurostat [45].”

Reviewer 1 suggested that we include a reference to a paper on the EHIS survey background and methodology by Hintzpeter et al. We would like to thank Reviewer 1 for bringing this reference to our attention as we were unaware of it. This is reference number 44 and to clarify for the reviewer, when we downloaded the citation, the listed author was the Robert Koch Institute. In order to complement this article, we have also included a reference to the EHIS methodological manual published by Eurostat. The text with both references on page 7 is as follows:

“Detailed information on the EHIS 2 methodology is available in a manual published by Eurostat [43] while a paper published by the Robert Koch Institute in Germany [44] provides a concise overview of the background and study methodology of the EHIS 2.”

Reviewer 1 suggested that we highlight the differences between the EHIS and other questionnaires such as EU-SILC or the Commonwealth Fund International Health Policy Survey. In order to address this comment, we have moved the following text from the Introduction section to the Data and Methods section on page 8:

“The EHIS differs from the EU-SILC and ESS as it does not ask respondents a binary (yes/no) question on whether they have unmet need. Rather, the EHIS asks respondents to consider unmet need arising from specific barriers to accessing health care, including long waits, distance or transportation problems and the affordability of services. The EHIS data allows the investigation of each component of unmet need separately and the consideration of financial barriers for medical, dental and mental health care and prescribed medicines.”

Reviewer 2 recommended that the Andersen model may help to expand the individual characteristics included in our study. We thank Reviewer 2 for this reference and agree this is a useful framework. Indeed, we consulted the framework and have included relevant variables insofar as the EHIS dataset allows us to.

Reviewers 1 and 3 asked that we clarify if the 4,004 respondents report a need for healthcare. Table A1 in Appendix 1 displayed the number and percentage of respondents who reported no unmet need, unmet need or no need for healthcare for each component of unmet need. We have moved this table to the main manuscript on page 22 so that it becomes Table 1. We also moved the first sentence of the Results section to page 8 to clarify that we exclude respondents who reported no need for healthcare. The revised text on reads as follows:

“Table 1 shows the number and percentage of respondents who reported no unmet need, an unmet need or no need for healthcare for each component of unmet need. The percentage of respondents who reported no need for healthcare ranged from 15% for affordability of dental care and prescribed medicines to 38% for mental health care.

Similarly, we moved the following text on missing data from the Results section to the Data and Methods section on page 9 so that it is clearer which observations were excluded from the initial database of 4,004 observations in order to reach the estimation samples:

“We excluded observations with missing data. Table A1 in Appendix 2 of the online supplementary material shows the percentage of missing data for each independent variable. Missing data for the explanatory variables did not exceed 7%, except for income, where 29% of observations had missing data. Table A2 in Appendix 2 shows the characteristics of respondents with missing income data. These were more likely female, aged 15-24, unmarried, of Luxembourgish nationality, had only primary and pre-primary education, a student, fulfilling domestic tasks or in compulsory service, had no limitations in activities due to a health problem and were less likely to be overweight or smoke daily.”

Reviewer 3 advised that we should mention if our models are (un)adjusted and include fixed or random effects. On page 9 we clarify that we include fixed effects (binary variables for the geographical level of canton) and report odds ratios adjusted for the explanatory variables. In the results section on page 10 and in the note for Table 4 on page 31 we also clarify that the results presented in Table 4 are adjusted odds ratios from the multivariate logistic regression models.

Reviewer 1 suggested we include the expression of the multivariate logistic regression function and we have included this on page 9:

As requested by the Editor we have created a separate section at the end of the Methods section on page 10 on Patient and Public Involvement:

“Patient and public involvement

Patients or the public were not involved in the design, conduct, reporting or dissemination of the EHIS.”

Results

As suggested by Reviewer 3 we revised the text describing Table 2 on page 10 so that it reads:

“Table 2 presents the number and weighted percentage of respondents according to the independent variables for each component of unmet need. A higher proportion of respondents who had bad or very bad health, a chronic disease, limitations in activities due to health problems or were ex-drinkers reported unmet need for all components. A higher proportion of respondents with obesity, low social support, and whose income fell below the first quintile reported unmet need due to distance and financial barriers. There was no discernible pattern between the remaining respondent characteristics and types of unmet need.”

Also in response to Reviewer 3, on page 11 we added text on the results of the sensitivity analysis to explain which groups were less likely to experience unmet need with respect to affordability of dental care:

“Table A3 in Appendix 2 of the supplementary data shows the results of the sensitivity analysis that included a binary variable measuring income non-response in each model. This variable was statistically significant in only the model for affordability of dental care with non-reporting of income associated with a lower risk of reporting unmet need due to the affordability of dental care (OR 0.64, 95% CI 0.42-0.97). Results for all models were largely unchanged. As in the main analyses, a lower risk of unmet need due to the affordability of dental care was associated with moderate (OR 0.43, 95% CI 0.24-0.78) or high (OR 0.40, 95% CI 0.22-0.72) social

support, income between the third and fourth quintile (OR 0.48, 95% CI 0.28-0.81) and fourth and fifth quintile (OR 0.30, 95% CI 0.17-0.54) and regular alcohol consumption (OR 0.60, 95% CI 0.42-0.87)."

Discussion

As advised by the Editor and Reviewer 3, we have included sub-headings in the Discussion section. In addition, we have added text to address the comments of all three reviewers.

On pages 11 and 12, we have added text to the section "*A statement of the principal findings*" providing an overview of our findings on the association of various determinants of unmet need varied according to the different components.

Reviewer 3 advised that we continue with comparisons to Belgium and France, rather than with other European countries. We agree that it would be preferable to compare Luxembourg to these two countries but this data is unavailable. We have deleted the text comparing Luxembourg to Italy on page 11 but we would like to keep the text on the European averages to provide some context for the Luxembourgish data.

Reviewer 1 suggested that we incorporate text on the impact of socioeconomic variables on waiting times. We have included the following text on page 13 that includes the reference provided and additional references for other relevant studies by L. Siciliani:

"Although we did not find any association between unmet need due to waits and socio-economic variables including education, income and job status, previous studies reported evidence of inequalities in waiting times related to education and income [58-60]."

In response to Reviewer 3, we have clarified our explanation of response and recall bias on page 13 as follows:

"As unmet need was self-reported, the data may suffer from the limitations inherent in survey data, including recall or response bias (respondent inaccurately remembers or misunderstands the question) [54, 55]."

We have redrafted the text on page 14 to clarify the definition of "collective households":

"Individuals living in collective households (for example, a boarding house or hostel or a dormitory in an educational establishment) [43] and institutions were also excluded."

We agree that some of the respondents with missing data may be younger people living in the family home, who do not bear the financial cost of care and have included the following text on page 14:

"The profile of respondents who did not report household income suggests that they were young people in full-time education who may have been living in the family home. Perhaps these respondents did not know the value of the household income, because they were not a main earner. We could assume that some of these respondents were less likely to bear the financial costs of care and therefore less likely to report unmet need due to the affordability of care. Indeed, the results of the sensitivity analysis as shown in Table A3 of Appendix 2 suggested this was the case, although the variable measuring income non-response is only statistically significant in the model measuring affordability of dental care."

We have responded to several comments from Reviewer 3 in the subsection "*The meaning of the study: possible explanations and implications for clinicians and policymakers*". We have moved the previous text on waiting times policies already implemented to the Introduction and deleted the comparison to waiting time targets in England. We included new text on pages 14-15 in the subsection "*Implications for clinicians and policymakers*" discussing policy responses that could be undertaken as a response to our findings. This addresses the question of Reviewer 3 if the results of

this paper could be used to target patients with specific characteristics that are most at-risk of unmet need.

Reviewer 3 also suggested that we highlight the limitations of the survey in terms of explaining waiting times for different types of services. We had already included text to this effect in the Discussion section and this text is now on pages 13 and 14 in the subsection “*Study limitations*”:

“The EHIS did not ask respondents if they perceived long waits across the health care system or for particular services (primary care, cancer care, elective treatment, diagnostic tests), nor the length of time perceived as long.”

Reviewer 2 suggested that social indicators could be expanded in subsequent revisions to EHIS and we have included the following text on page 15 in the subsection “*Implications for future research*”:

“Future revisions of the EHIS could expand the social variables collected, for example by adding information on ethnicity or parents’ country of birth. Additional questions on social support could help to distinguish between emotional, instrumental and informational support [19].”

Conclusions

The editor asked that we provide a separate conclusions section in the main manuscript that summarises the findings on unmet need for healthcare in Luxembourg. We have drafted a new Conclusions section on page 16 as follows:

“Compared to other EU countries, Luxembourg has high per capita public health care expenditures, a low share of total health spending financed by out-of-pocket payments and a comprehensive benefit package [61, 62]. Nevertheless, this study provides evidence that some individuals experienced difficulties accessing care, particularly due to long waits and affordability of care. Future reforms to improve access to health care should first target high-risk groups including those with low incomes and poor health. Future policies to address unmet need should also consider other vulnerable populations who lack formal health care coverage including the homeless, undocumented immigrants and those in the informal labour market.”

In addition, we have responded to the two additional comments by Reviewer 3:

Table 1 – does the percentage refer to the weighted percentage of patients who reported unmet need for each individual component? It may be worth clarifying this in the note.

We have clarified in the note for Table 2 that the percentage is for each component. We have also added a column on the number of respondents who reported unmet need for each component as we realised that the numerator was not clear. We also removed the N and % for “Any unmet need” and “Any affordability” and the corresponding text in order to maintain consistency with Tables 1, 3 and 4 where we only report the individual components of unmet need.

The authors should try and write the paper in past tense. Most of the paper is written in the present tense.

We agree that the Data and Methods and Results sections should be written primarily in the past tense and we have amended these sections accordingly. In the Introduction, we have written the review of previous literature in the past tense but feel it is appropriate to write the subsection on the Luxembourgish health system in the present tense and also the first paragraph of the Introduction. Similarly, in the Discussion, we have written the sections summarising the principal findings and comparing our results with previous studies in the past tense but the “Implications for clinicians and policymakers” subsection in the present tense.

Individual comments and page numbers:

Editor's Comments to Author (if any):

- Can you please state more clearly what your objectives are at the end of the introduction section? What is your research question? We currently find the purpose of this study a little confusing.

Introduction, The Luxembourg health system and access to health care: page 7

The conclusions are exclusively focused on the usefulness of the EHIS rather than the data it provides on unmet need for health care in Luxembourg. Can you modify the conclusions section of the abstract so that it summarises what the findings are on unmet need for health care in Luxembourg?

Abstract: page 2

Please also provide a separate conclusions section in the main manuscript. We note that reviewer 3 (below) raises similar concerns.

Conclusions: page 16

- Methods: "Patients or the public were not involved in the design, conduct, reporting or dissemination of the EHIS." Can you please move this statement to a separate section at the end of the methods with the sub-heading 'Patient and Public Involvement'?

Data and methods, Patient and Public Involvement: page 10

- Can you clarify what you mean by the 'participation rate' on page 6? Is this the same as the completion rate or the response rate? If the former then what was the response rate for the survey?

Data and methods, Study population and design: page 8

- Please work on improving the discussion section. For example, we can't locate an in-depth discussion of the study's strengths and limitations. This section should broadly cover the following areas: a statement of the principal findings; strengths and weaknesses of the study; strengths and weaknesses in relation to other studies, discussing important differences in results; the meaning of the study: possible explanations and implications for clinicians and policymakers; and unanswered questions and future research. We suggest using sub-headings to guide the reader.

Discussion: pages 11-15

Reviewer Reports:

Reviewer: 1

Dr. Francisco Reyes-Santías, Universidad de Vigo, Servicio Galego de Saude

Comments to the Author:

The paper "Investigating unmet need for health care using the European Health Interview Survey: a cross-sectional survey study of Luxembourg", studies the determinants of the unmet demand for healthcare through the information contained in the European Health Interview Survey (EHIS) for Luxembourg.

Title and summary. The title and abstract express well the object of study, objectives and results of the article.

Structure of the article. The contents are well organized and they adhere to the IMRaD structure. It includes a theoretical framework of the research problem but at this point I suggest the authors incorporate a bibliographic reference that I miss in the text:

Hintzpeter B, Finger JD, Allen J (2019) European Health Interview Survey (EHIS) 2 – Background and study methodology. *Journal of Health Monitoring* 4(4):66-79.

Data and methods, Study population and design: page 7

Focusing on the opportunity of the study, it must be said that it is a useful work. Because the paper studies the possible determinants of difficulties in accessibility and affordability, taking into account sociodemographic characteristics of the population under study, lifestyle habits and characteristics of the health system such as copayments or waiting lists. In that sense, hypotheses could be established in a clearer way for the reader than as they are intended to be established (i.e. "unmet needs vary according to services" -what type of services: surgical, hospitalization, or primary care, etc. -) (page 4, lines 47-52).

Introduction, The Luxembourg health system and access to health care: page 7

Materials and methods.

Regarding the material and methods section, the methodology is tailored to the object of study and the objectives and is explained in a transparent manner while it has been validly applied to guarantee the results.

However, I suggest that the authors clarify some issues in the text:

1.- It would be interesting if authors indicated that the 4000 subjects in the sample refer to people who have not shown a need for health care and those who have. otherwise, the exclusion of people who have not required healthcare would lead to higher ratios of unmet needs for the rest of the population.

Data and methods, Study population and design: page 8 and Table 1: page 22

2.- Authors could also indicate for readers unfamiliar with the EHIS questionnaire that, unlike other questionnaires such as the EU-SILC or the Commonwealth Fund International Health Policy Survey, it directly asks all respondents about the barriers they have encountered in receiving care. not if they needed such care in the first place.

Data and methods, Study population and design: page 8

3.- Authors are suggested to incorporate the expression of the multivariate logistic regression function that they use to analyze unmet needs due to waiting times, distance and affordability.

Data and methods, Statistical data analysis: page 9

Results.

The results are significant and they are presented in an adequate and understandable way not only through narration, but also with self-explained tables and figures that are also well elaborated in terms of presentation. The results justify and relate to the objectives and methods and the results are of sufficient social interest.

Discussion.

The discussion appropriately compares the study results with other works, highlighting the main study findings. 75% of the bibliography cited in the study belongs to the previous five years. But it is suggested to the authors that the discussion incorporates the topic of the impact of socioeconomic variables on the waiting time to access health care; such as the study by Simonsen et al:

Simonsen NF, Oxholm AS, Kristensen SR, Siciliani L. What explains differences in waiting times for health care across socioeconomic status? *Health Econ.* 2020 Dec;29(12):1764-1785. doi: 10.1002/hec.4163. Epub 2020 Sep 29. PMID: 32996212.

Discussion, comparisons with previous studies: page 13

The conclusions are adequately related to the objective.

Overall, it is an interesting study, and should be considered for publication in *BMJ Open*, once the minor revisions proposed have been resolved.

Reviewer: 2
Dr. Justine De Oliveira, Duke University

Comments to the Author:

Well described research strategy, well written manuscript, thoughtful discussion of findings. A useful framework for comparison of access to care across EU countries. Social indicators could be expanded in subsequent revisions to EHIS.

Discussion, Implications for future research: page 15

While focused on the US, the Andersen model (Aday LA, Andersen R. A framework for the study of access to medical care. Health Serv Res. 1974;9(3):208–220.) and its subsequent revisions may offer food for thought, and expanding consideration of predisposing factors and patient satisfaction and their impact on access to care.

We have consulted the Andersen model and included variables insofar as they are available in the EHIS.

Reviewer: 3
Miss Lucina Rolewicz, Nuffield Trust

Comments to the Author:

In general this is a well-written paper, and I think it will add to the landscape of literature in this area.

However, I have some detailed comments about the structure and the relevance of some of the content, which I have attached. I would recommend that the authors consider incorporating these suggestions to strengthen the key messages that the paper is trying to get across.

Abstract

Objectives – this is slightly unclear. Is the aim of the paper to “illustrate the value of the EHIS” or rather is it to investigate the barriers most associated with patient-reported unmet need in healthcare, and the relationship between patient characteristics and the components of unmet need? If the former, then the results should reflect how “value of the EHIS” is measured. It could also be more concise – can mention that this focuses on Luxembourgish patients (or patients residing in Luxembourg) in a preceding sentence and exclude the last part of line 7 (page 3) “...as an interesting and policy-relevant case study.”

Design – statistical methods are not relevant to the study design and can be omitted.

Settings and participants – are the participants registered with health care services in Luxembourg? This could be clarified in the text.

Conclusions – again, this doesn’t seem to be the main aim of the paper – I would say the breadth of the unmet need definition is just a strength of the EHIS survey. The results are showing which patient group (i.e. those with poor health) is associated with the highest levels of unmet need. It is not clear how the usefulness of the EHIS in measuring unmet need is a novel finding, as it has been used previously, as outlined on page 5, line 25. I understand that previous studies have used the survey to analyse specific sub-groups, but generally using the EHIS as a measure of unmet need has been done before. The conclusion should set out how the main results can help address unmet need amongst patients in Luxembourg.

Abstract: page 2

Introduction

In general, there is a lot of unnecessary detail in the introduction. The introduction could benefit from a clearer structure on, for example, why is measuring unmet need important; the literature demonstrating unmet need in healthcare and the implications unmet need has on patients/the system, specifically in Luxembourg; where unmet need arises from (i.e. wait times, affordability, distance/transportation, any other factors beyond the scope of the survey?), noting any variation between patient groups and healthcare services; and specific ways that government and/or health services are trying to address the different aspects of unmet need. This should touch on areas such as policies aimed at improving waiting times (which are mentioned in the discussion section), seeing

as it is a medium-high priority in Luxembourg and is an indicator of unmet need, and distance/transportation (if policies exist e.g. digital transformation of care).

Page 4, lines 22-52 could be written more concisely. Sentences on lines 34-42 could be reworded and possibly removed. This paragraph would also sit better at the end of the introduction section as it will flow better into the study methods.

Page 4, line 45 “potentially more thorough way” sounds too doubtful, would consider removing “potentially”.

Page 4, lines 47-52 – it isn’t entirely clear why Luxembourg is interesting relative to other countries. The authors mention different cost barriers, but it isn’t clear what these distinct costs are.

Page 5, lines 3-45 – this level of detail about funding is not necessary, given the stated aim of the paper. Some background is useful, but the important point here is how funding mechanisms described relate to unmet need in patients (page 6 line 47 – page 7 line 13). Belgium and France are mentioned as examples where some population groups have their costs covered. It is more important to explicitly mention if this is the case in Luxembourg – or is the cost of healthcare the same for all patients? The authors have mentioned in the next paragraph that those in economic hardship can request assistance with payments, but this would benefit from being mentioned in the preceding paragraph as a comparison with Belgium and France, and how this helps (if at all) address unmet need. Also, it is worth clarifying whether the universal third-party payment system replaces the third-party social payment.

Introduction: pages 4-7

Data and methods

Page 6, line 53 – it’s not entirely clear from the text if 4,004 individuals reported a need for care in the survey. If not, another value representing the number of respondents indicating a need for care would be useful here.

Data and methods, Study population and design: page 8 and Table 1: page 22

Page 7, lines 28-38 – the authors should mention whether these models are unadjusted or adjusted and do they include fixed effects or random effects?

Data and methods, Statistical data analysis: page 9 and Table 4, page 31

Results

Page 8, lines 22-32 – should the wording here be “A higher proportion of respondents who had bad or very bad health, a chronic disease, limitations in activities due to health problems or were exdrinkers reported unmet need for all components” as it relates to frequencies and percentages? The same applies for the following sentence in the paragraph.

Results: page 10

Page 9, lines 30-35 – which group were less likely to experience unmet need with respect to affordability of dental care? This should be explicitly stated in the text.

Results: page 11

Discussion

It is unclear how comparisons with European countries (not previously mentioned in the paper) relate to discussing the main findings of the study. Since the introduction drew on comparisons between Luxembourg and France/Belgium, it may be better (if possible) to continue with this comparison into the discussion.

Data on France and Belgium is not available

Some reorganisation of this section may benefit the reader. For example, starting with reiterating the main results; followed by comparisons with previous studies and strengths and limitations of this study; followed by implications for policy and clinical practice; and finishing with implications for future research.

Discussion: Pages 11-15

Could the results of this paper be used to target patients with specific characteristics that are most at-risk of unmet need?

Discussion, Implications for clinicians and policymakers: Page 15 and Conclusions: Page 16

Page 10, line 12 – the part of the sentence in brackets doesn't seem to make sense – would consider revising.

Discussion, Study limitations: Page 13

Page 10, line 15 – health is spelled incorrectly.

Discussion, Study limitations: Page 13

Page 10, line 20 – the authors mention that individuals living in collective households were excluded. This could be my misunderstanding, but would this exclude young people (e.g. aged 15-18) who live in the family home? If the study includes these patients, is the finding that a higher proportion of the younger age group have missing income data an expected finding, since they are less likely to have an income and are also less likely to bear the financial cost of care? If either of these points are true, this would be worth clarifying in the text.

Discussion, Study limitations: Page 14

Page 10 – one limitation of the results is that they cannot compare unmet need arising from waiting times across different types of healthcare services. Waiting times from different services are mentioned in detail in the discussion so the authors should highlight the limitations in how far the study data is able to explain these differences.

Discussion, Study limitations: Pages 13 and 14

Page 12, lines 19-20 – I am not sure how helpful it is to compare waiting targets between Luxembourg and England – I would imagine the population needs are very different, as one example.

Sentence deleted

Other comments

Table 1 – does the percentage refer to the weighted percentage of patients who reported unmet need for each individual component? It may be worth clarifying this in the note.

Table 2: page 23

The authors should try and write the paper in past tense. Most of the paper is written in the present tense.

VERSION 2 – REVIEW

REVIEWER	Francisco Reyes-Santías Universidad de Vigo, Organización de Empresas e Mercadotecnia
REVIEW RETURNED	06-Jun-2021
GENERAL COMMENTS	Dear Editor, Regarding the review of paper "Investigating unmet need for health care using the European Health Interview Survey: a cross-sectional survey study of Luxembourg" (bmjopen-2021-048860.R1), once I have read the letter of reply from the authors to my comments on the

	review, I understand that they have covered all my suggestions and, as far as I am concerned, my suggestion to BMJ Open is to accept the publication of the new version of the paper.
--	---